# Response of Cladocera Fauna to Heavy Metal Pollution, Based on Sediments from Subsidence Ponds Downstream of a Mine Discharge (S. Poland)

**Agnieszka Pociecha [1],\*** , **Agata Z. Wojtal [1]**, **Ewa Szarek-Gwiazda [1]**, **Anna Cieplok [2]**, **Dariusz Ciszewski [3] and Andrzej Kownacki [1]**

[1] Department of Freshwater Biology, Institute of Nature Conservation, Polish Academy of Sciences, Adama Mickiewicza Av. 33, 31-120 Kraków, Poland; wojtal@iop.krakow.pl (A.Z.W.); szarek@iop.krakow.pl (E.S.-G.); kownacki@iop.krakow.pl (A.K.)

[2] Department of Hydrobiology, University of Silesia, Bankowa St. 9, 40-007 Katowice, Poland; anna.cieplok@us.edu.pl

[3] Faculty of Geology, Geophysics and Environmental Protection, University of Science and Technology AGH, Adama Mickiewicza Av. 30, 30-059 Kraków, Poland; ciszewski@geol.agh.edu.pl

\* Correspondence: pociecha@iop.krakow.pl

**Abstract:** Mining is recognized to deeply influence invertebrate assemblages in aquatic systems, but different invertebrates respond in different ways to mining cessation. Here, we document the response of the cladoceran assemblage of the Chechło river, S. Poland (southern Poland) to the cessation of Pb-Zn ore mining. The aquatic system includes the river and associated subsidence ponds in the valley. Some ponds were contaminated during the period of mining, which ceased in 2009, while one of the ponds only appeared after mining had stopped. We used Cladocera to reveal how the cessation of mine water discharge reflected on the structure and density of organisms. A total of 20 Cladocera taxa were identified in the sediment of subsidence ponds. Their density ranged from 0 to 109 ind./1 $cm^3$. The concentrations of Zn, Cd, Cu and Pb were much higher in sediments of the ponds formed during peak mining than in the ponds formed after the closure of the mine. Statistical analysis (CCA) showed that *Alonella nana*, *Alona affinis*, *Alona* sp. and *Pleuroxus* sp. strongly correlated with pond age and did not tolerate high concentrations of heavy metals (Cu and Cd). This analysis indicated that the rate of water exchange by the river flow and the presence of aquatic plants, affect species composition more than pond age itself.

**Keywords:** Zn-Pb maine; subfossil; Cladocera; heavy metals; CCA analyses; anthropogenic impact

## 1. Introduction

Cladocera (Crustacea) are an important component of the small invertebrates living in freshwater. They occur in different environmental conditions; in shallow and deep waters; in alkaline, neutral and acidic conditions, and among environmental gradients. Their sensitivity to environmental conditions makes cladocerans good indicators for a wide range of environmental variables [1–3]. They respond rapidly to heavy metals and to other physical-chemical variables affected by a discharge of mine waters [1,3,4]. These environmental changes are recorded in bottom sediments and may be reconstructed from subfossil skeletal remains, preserved in the bottom layer of mud. For this reason, they are useful for the reconstruction of anthropogenic disturbances [4–7].

Extensive investigations conducted at mining sites throughout the world have recognized metal-contaminated wastes as the most important source of heavy metal contaminants. Lead and zinc (Pb, Zn) ores have been extracted for centuries in mines all over the world. In the second half of 20th

century many mines have been closed, due to ore exhaustion, but leaving contaminated sediments as sources of metals leaking to the aquatic environment. The Chechło River valley in S. Poland is an example of an aquatic system affected by Pb-Zn mining, in which sediments of mining-related subsidence ponds are contaminated with heavy metals (Pb, Zn, Cd). For about 40 years, until 2009, the Chechło River has experienced pollution from a mine active since 1968. The lower reach of the Chechło River is affected by longwall coal mining. This resulted in subsidence of the river valley floor and emerging of two basins ponded with water. One of the basins started to subside at the beginning of the 1990s whereas subsidence of the other, 1 km upstream, started at about 2007. In both basins subsidence reached about 1.7–2.0 m resulting in shallow water bodies with plant succession much more advanced in the older than in the youngest basin [8]. Currently, the river is recovering, but many channels and floodplain locations still preserve the sediments that accumulated there during the mining time [8]. This allows the assessment of the impact of contaminated waters of the river and ponds on Cladocera communities based on the analysis of their remains in the sediments.

Our aim was to document changes in the species composition of Cladocera in response to mining cessation recorded in bottom sediments of subsidence ponds situated on the Chechło River floodplain (southern Poland). Our hypothesis assumes that regeneration of the cladoceran community was limited by high heavy metal concentrations (mainly heavy metals, such as: Cd, Pb, Zn and Cu) in the bottom sediments of ponds in the middle reach of the river valley. We analyzed the Cladocera assemblage and heavy metal concentrations in sediment cores from ponds which emerged (1) during the period of Zn and Pb ore extraction and (2) after the mine closed. The present study is the key to understanding the rate of ecosystem response and factors controlling ecosystem recovery from heavy metal contamination. We address this by analyzing Cladocera remains preserved in subsidence ponds sediment and correlate them with records of metal contamination from Pb-Zn ore mining.

## 2. Materials and Methods

### 2.1. Study Area

Our work was conducted in the middle course of the Chechło River, downstream of the point of mine waters discharge. The quality of the Chechło River was affected for about 50 years by the zinc and lead ore mining and by the other industrial and municipal sewage effluents from two towns Trzebinia and Chrzanów located in the middle reach of the river [8,9]. We distinguished two research areas about 1 km apart: Subsidence pond emerged after the closure of the mine (UP) and subsidence ponds ponded during the peak of the ore exploitation (DOWN) (Figure 1). All water bodies have been impacted by heavy metals contamination. They are small, with areas ranging from 0.5 to ca. 5 ha, whereas their average depth is about 1–2 meters. A part (ca. 20–50%) of the basins is overgrown with macrophytes (Figure 1).

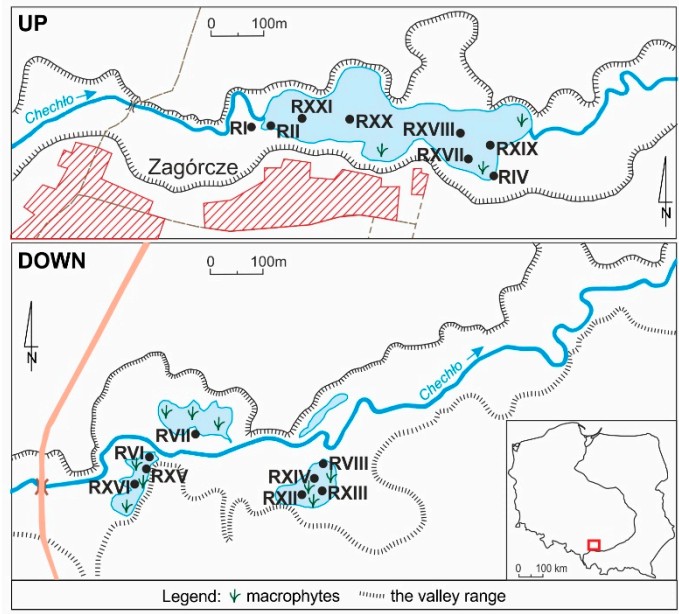

**Figure 1.** Sampling area (UP: Subsidence pond after the closure of the mine; cores: RI, RII, RIV, RXVII, RXVIII, RXIX, RXX, RXXI; DOWN: Subsidence ponds during peak exploitation; cores: RVI, RVII, RVIII, RXII, RXIII, RXIV, RXV, RXVI).

## 2.2. Sampling and Measurements

The concentrations of Cd, Pb, Zn and Cu and remains of aquatic organisms were analyzed in sediment cores from eight sites in pond formed after the closure of the mine (UP, cores: RI, RII, RIV, RXVII, RXVIII, RXIX, RXX, RXXI) and in eight sites of ponds existing during the peak of Zn and Pb ore exploitation (DOWN, cores: RVI, RVII, RVIII, RXII, RXIII, RXIV, RXV, RXVI) (in total 16 cores) (Figure 1). Cores were recovered using a Multisampler piston corer with diameter 4.5 cm (Eijkelkamp, Giesbeek, Netherlands). Cores were divided in the field depending on sediment lithology when macroscopic changes in color or grain-size of sediments were observed whereas profiles with no distinct strata were divided into 10 cm long subsamples and homogenized by mixing. Length of the retrieved profiles varied from 15 to 70 cm. Most of the sediments were composed of unstratified muds with a high content of organic matter which rested on the sandy sediments of the former floodplain. UP and DOWN pond cores were divided into sections with a thickness of 5–15 cm depending on the lithology of the sediment (see Figure 2).

Sediment samples for heavy metal analysis were dried at 105 °C and sieved through a 0.063 mm sieve. Then they (0.5 g) were digested with 10 cm$^3$ of 65% HNO$_3$ and 2 cm$^3$ of 30% H$_2$O$_2$ (both analytical grade) using a microwave digestion technique [6]. The Cd, Pb, Zn and Cu concentrations were determined using a flame atomic absorption spectrometer (F-AAS). Metal analyses were performed according to (standard certified) analytical quality control procedures.

Subfossil Cladocera preparation was conducted, according to Frey [5]. One centimeter cubed of fresh homogenized sediment was taken from the separation section from each core for cladoceran analysis. After elimination of carbonates using HCl, each sample was boiled for half an hour in a 10% KOH solution. A magnetic stirrer was used for dispersion. After boiling and washing, the remains were sieved through a 35 μm sieve. The residuum was stored in 10 ml of water with glycerine and safranine, in polycarbonate test-tubes in a fridge. Temporary slides were used for identification and to count the cladoceran remains. Taxa were identified and counted at 200 or 400× magnification under a Nikon 50i microscope. All skeletal parts were counted: headshields, shells, postabdomens, postabdominal claws, ephippia and filtering combs. The most abundant body part for each taxon was chosen to represent the number of individuals. The results of qualitative and quantitative analyses are presented in diagrams,

in which an absolute number of specimens was calculated for 1 cm$^3$ sediment volume. Identification of the species was based on Frey [10] and Szeroczyńska and Sarmaja-Korjonen [11].

*2.3. Statistical Analyses*

Shannon's diversity index

The species diversity of Cladocera was evaluated using Shannon's diversity index. The analysis of species diversity was carried out using the MultiVariate Statistical Package (MVSP) 3.1 program [12].

Pearson's correlation

In order to investigate the relationship between heavy metals content in sediments and cladocerans, Pearson's correlation was used with the Statistica 13 program.

Cluster analysis

In order to compare qualitatively the Cladocera assemblages cluster analysis was made using the UPGMA method. The hierarchical classification was obtained using the MVSP 3.1 program [12].

Correspondence analysis

Indirect and direct ordination techniques were used to assess the impact of sediment environmental factors on the cladoceran community: Unimodal techniques—correspondence analysis (CA) and canonical correspondence analysis (CCA).

The analysis was carried out after a prior detrended correspondence analysis (DCA) to verify the nature of the data, based on the length of the gradient expressed in standard deviation (SD) units. During the DCA, CA and CCA analysis, the data was log-transformed [ln (x + 1)] and centered. During the CCA analysis, a forward selection was carried out to assess the role of environmental variables in shaping the structure of cladoceran communities. The used analyses were performed on Cladocera data and subsidence sediments samples to identify the changes in the ponds and to show the relationships between the environmental variables and the distribution of the Cladocera.

An evaluation of their statistical significance, as well as the statistical significance of canonical axes, was made using the Monte Carlo permutation test for 499 repetitions. The analyses were performed using CANOCO for Windows 4.5 program [13].

## 3. Results

*3.1. Sediment—Heavy Metals Contamination Analysis*

The heavy metal concentrations in sediment cores from studied UP and DOWN subsidence ponds ranged as follows: Zn, 0.5–23.1 mg g$^{-1}$; Cd, 6.1–612.3 µm g$^{-1}$; Pb, 0.3–10.2 mg g$^{-1}$; Cu, 21.4–397.0 µm g$^{-1}$ (Figure 2). The sediment from the UP subsidence pond exhibited lower differences in heavy metal concentrations among particular strata of each individual core (with the exception of the core RXVIII). Only in core RXVIII (length 20 cm) collected from the lower part of the UP subsidence pond, they were considerable with maximum concentrations of the metals of all cores in UP subsidence pond were found in the bottom strata. The cores from DOWN subsidence ponds, usually of a larger length, exhibited higher variability in heavy metal concentrations which each core, comparing to UP subsidence pond. These differences were particularly large (up to 24 times for Zn, 19 times for Cd, 13 times for Pb, and 7 times for Cu) in 30 cm long cores RVIII and RXII. In the deepest section of core RXII the lowest metal concentrations for all DOWN subsidence ponds were found. Higher metal concentrations usually occurred in the middle or the surface strata of cores from DOWN subsidence ponds and reflected the period of most intensive mining of Zn and Pb ores at the end of the 20th century. The maximum concentrations of heavy metals in cores from DOWN subsidence ponds were several times higher (Zn, Cu and Pb 2–4, Cd eight times) than from the UP subsidence pond (with the exception of RXVIII core).

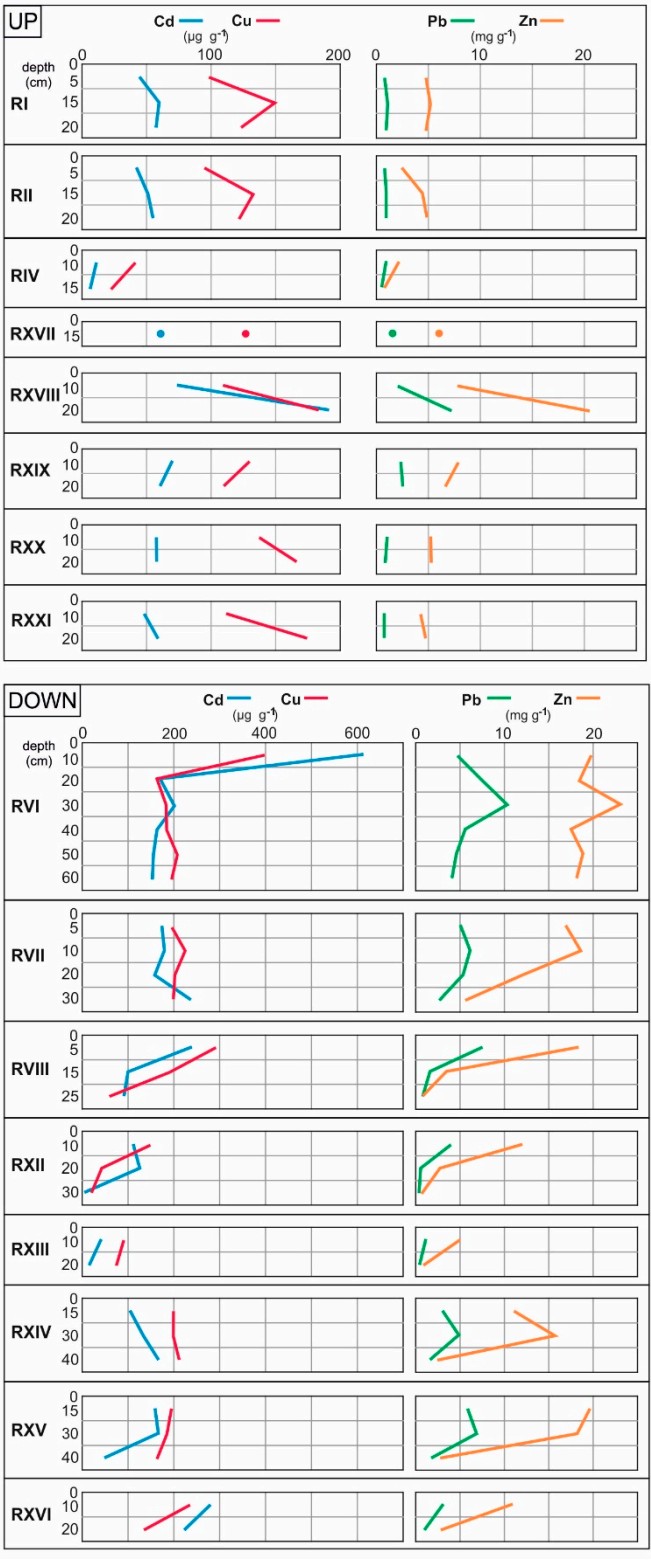

**Figure 2.** Heavy metal concentrations in the sediment cores in subsidence ponds (UP: Subsidence pond after the closure of the mine; cores: RI, RII, RIV, RXVII, RXVIII, RXIX, RXX, RXXI; DOWN: Subsidence ponds during peak exploitation; cores: RVI, RVII, RVIII, RXII, RXIII, RXIV, RXV, RXVI).

### 3.2. Subfossil Cladocera Analysis

Cladoceran remains in the sediments of the subsidence ponds represented 20 taxa and four families (Chydoridae, Eurycercidae, Bosminidae, Daphniidae). Chydoridae was the most diverse

family (13 taxa), whereas the other families were represented by fewer taxa (Eurycercidae: 1; Bosminidae: 2; Daphniidae: 4). In the pond which emerged after the mine was closed (UP) 12 taxa were found while in ponds existing during the mine operation (DOWN) 17 taxa were found. The number of cladoceran taxa in each sediment core ranged from 1 to 13 (Table 1).

**Table 1.** Number of taxa and range (min.–max.) of total Cladocera specimens in cores of subsidence ponds.

| Subsidence Ponds Area | Core Number | Number of Taxa in Core | Range (Min.–Max.) of Cladocera Individuals in 1 cm$^3$ of Sediment |
|---|---|---|---|
| UP (subsidence pond formed after the closure of the mine) | RI | 5 | 1–6 |
| | RII | 3 | 0–2 |
| | RIV | 7 | 4–22 |
| | RXVII | 5 | 37 |
| | RXVIII | 3 | 1–12 |
| | RXIX | 4 | 0–15 |
| | RXX | 5 | 3–6 |
| | RXXI | 1 | 1 |
| DOWN (subsidence ponds created during the peak of Zn and Pb ore exploitation) | RVI | 3 | 3–8 |
| | RVII | 2 | 1–6 |
| | RVIII | 6 | 1–50 |
| | RXII | 13 | 23–109 |
| | RXIII | 3 | 22–101 |
| | RXIV | 6 | 38–86 |
| | RXV | 6 | 20–93 |
| | RXVI | 4 | 3–76 |

The pond existing during the mine operation had the highest diversity of Cladocera taxa in comparison to remaining subsidence ponds. Total density of Cladocera individuals in sediments (ind./1 cm$^3$) varied from 0 in UP subsidence pond formed after the closure of the mine (RII; RXIX) to over 100 individuals in DOWN ponds existed during mine exploitation (RXII; RXIII) (Table 1). In the UP pond Cladocera assemblage was rather poor and its density was not higher than 37 ind./1 cm$^3$. This result is probably related to the fact that in subsidence ponds where sediment cores were studied the river water flowed through the center of the pond (Figure 1). The DOWN subsidence ponds are supplied with river water by side channels, and characterized by stable water with well-developed macrophyte vegetation (Figure 1). *Chydorus sphaericus* was the species present in all studied ponds except one, with the presence of the only one species *Bosmina longirostris* (RXXI; Figure 3). *Ch. sphaericus* is species occurring in pelagic and littoral zones, and its high density is characteristic for eutrophic and polluted water. The highest densities of this species were observed mainly in the lowest layers of core sediments with more than 40 ind./1 cm$^3$ in subsidence ponds existed during the peak of Zn and Pb ore exploitation (DOWN). The smaller density of the discussed species, below 10 ind./ 1 cm$^3$ were observed in cores of the pond formed after the mine was closed (UP) (Figures 3 and 4).

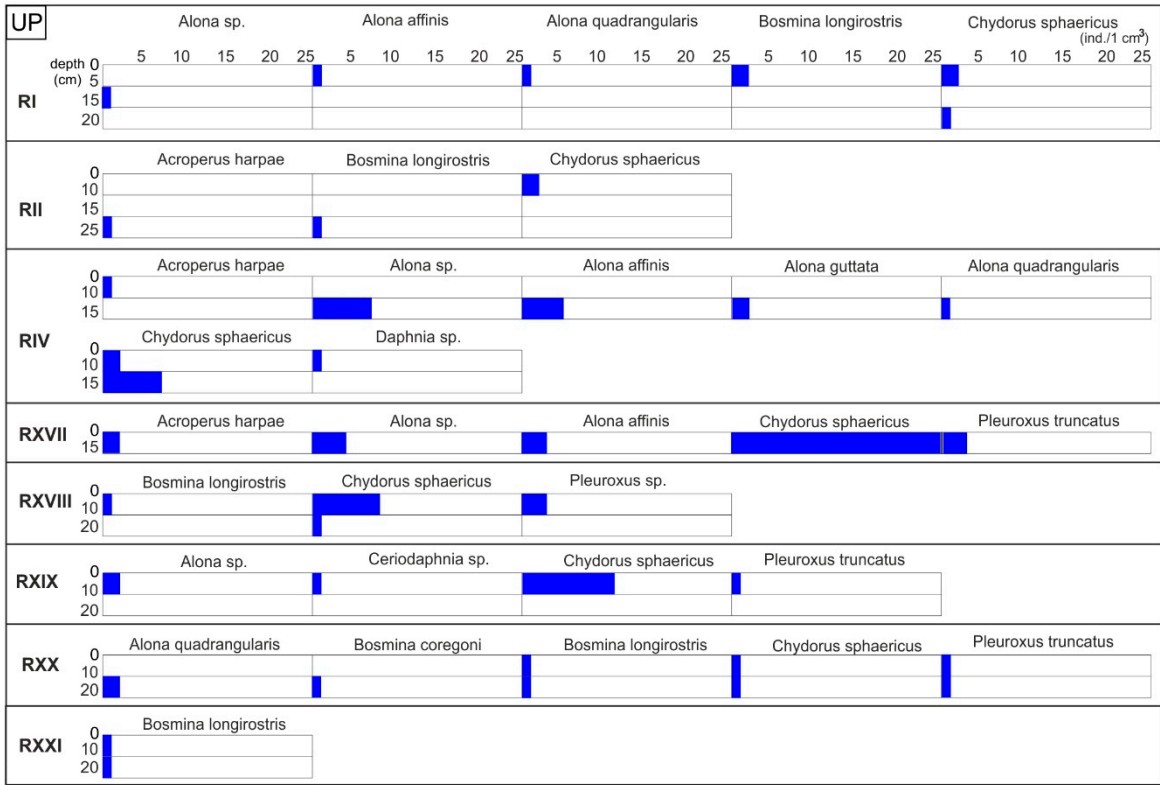

**Figure 3.** Diagram of the absolute number of Cladocera individuals in 1 cm³ of sediment from subsidence pond formed after the closure of the mine (UP).

We found seven littoral taxa in the UP pond characterized by through-flow of river water and 12 littoral taxa in DOWN ponds characterized by stagnant water conditions and littoral zone overgrown with macrophytes. In all ponds were found littoral taxa belonging to six genera: *Alona*, *Alonella*, *Acroperus*, *Pleuroxus*, *Eurycercus* and *Graptoleberis*. Only in one subsidence pond was the highest number of littoral taxa (9 taxa—RXII: DOWN) with density reaching over 30 ind./1 cm³ (Figure 3).

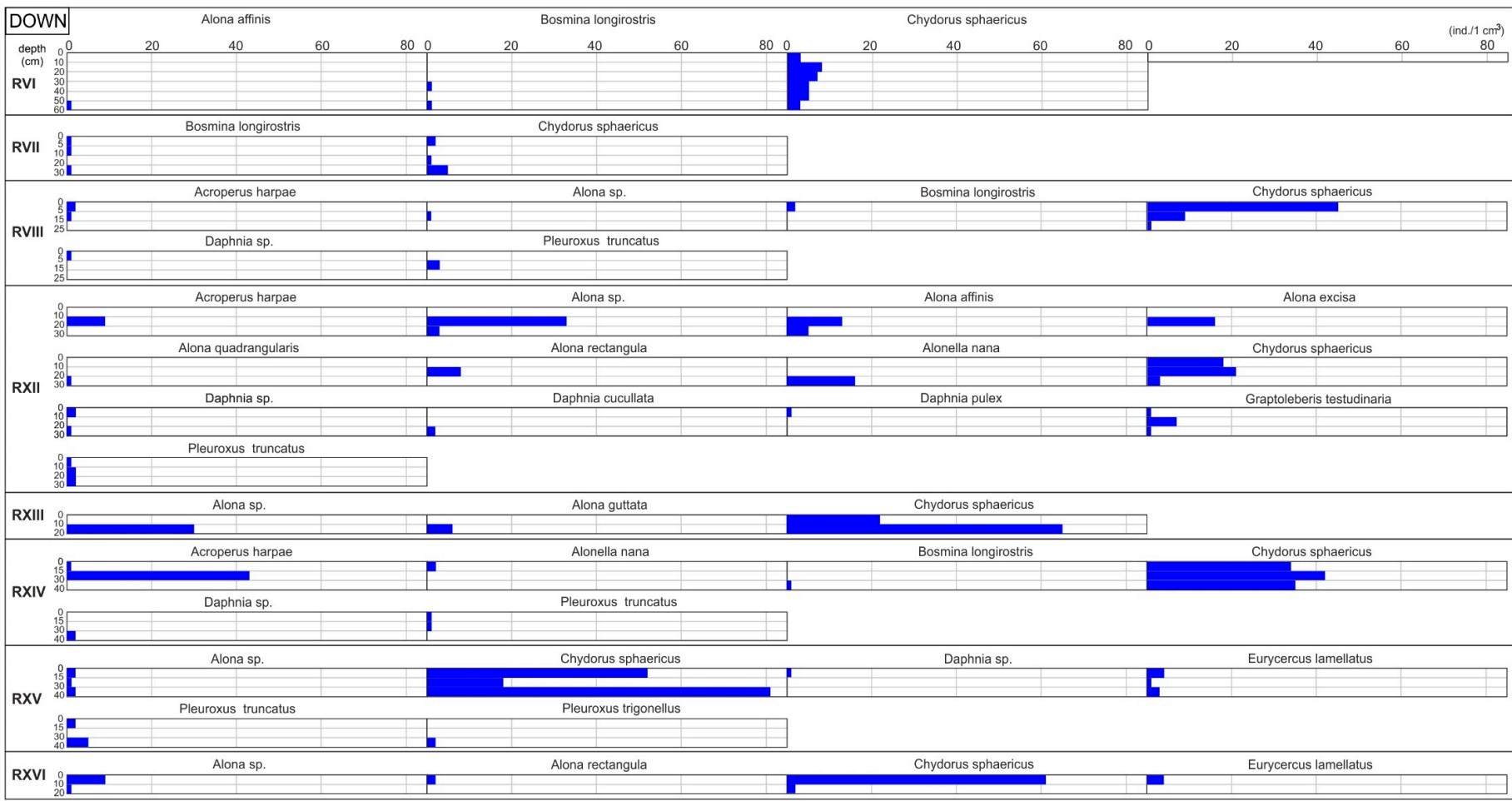

**Figure 4.** Diagram of the absolute number of Cladocera individuals in 1 cm³ of sediment from subsidence ponds existed during mine exploitation (DOWN).

### 3.3. Relationship between Cladocera and Heavy Metal Concentration

The highest Shannon (H′) diversity rates (up to <2) occurred only in one core RXII in the subsidence pond formed during mine operation (DOWN). Moreover, the high H′ diversity rates were observed in five cores (up to <1) in the water ponds formed after of mine cessation (UP) (Table 2).

**Table 2.** Shannon (H′) diversity index value in investigated sediment cores. (UP—subsidence ponds formed after the closure of the mine; DOWN—subsidence ponds created during the peak of Zn and Pb ore exploitation).

| Core (UP) | Index Value | Core (DOWN) | Index Value |
|---|---|---|---|
| RI | 1.49 | RVI | 0.35 |
| RII | 1.04 | RVII | 0.58 |
| RIV | 1.61 | RVIII | 0.66 |
| RXVII | 1.07 | RXII | 2.12 |
| RXVIII | 0.79 | RXIII | 0.73 |
| RXIX | 0.85 | RXIV | 0.80 |
| RXX | 1.58 | RXV | 0.57 |
| RXXI | 0.00 | RXVI | 0.68 |

There were statistically significant correlations between the abundance of particular species and the heavy metal concentrations in sediment cores of UP and DOWN ponds. In the most contaminated sediments (DOWN ponds), a negative correllation was found between Cu concentration and occurrence of the genera *Alona*, *Alonella*, *Daphnia* and *Graptoleberis*; also, a negative correlation was revealed between the density of *Alona* sp. and Zn and Pb in sediments. Negative correlations with Cu concentration was found in less polluted sediments (UP pond), but only for *Alona* and *Daphnia* genera. The cladoceran *Alona* and *Daphnia* seem to be more sensitive to heavy metals contamination than the other investigated Cladocera (Table 3). Our results showed the negative impact of metal mining on cladocerans, as well as positive correlations between taxa.

Cluster analysis divided the studied cores into two groups. This was mainly caused by the different dominance structure of the identified species of Cladocera at sampling sites. The first group constitutes cores from ponds that existed during the mine exploitation activity; the second group included water bodies created after the mine was closed. In the first group, site RXII clearly stood out, due to the fact that it was characterised by the highest density and variety of Cladocera, as well as the highest values of Shannon's diversity index (H′ = 2.12). In the second group apart from cores from UP pond, also cores RVI and RVII from DOWN ponds were included (Figure 5). Such classification was likely to be related to the small diversity and density of Cladocera and low values of Shannon's diversity index (H′ = 0.35 and 0.58 respectively) in the cores RVI and RVII.

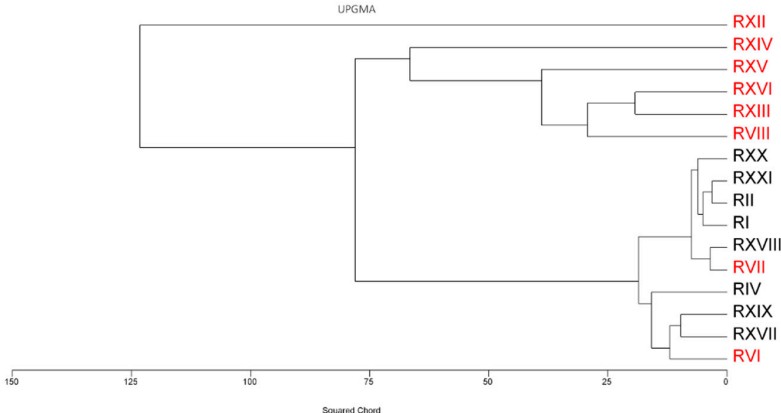

**Figure 5.** Dendrogram of faunistic similarities for Cladocera fauna in the studied sediment cores (black color: Cores from pond, created after mine was closed (UP); red color: Cores from ponds existed during mine exploitation (DOWN)).

To show the statistical importance and the associations of Cladocera assemblages with particular types of ponds correspondence analysis (CA) was used. The results showed that the first axis explains 25.4%, while the second 17% of the variability of the occurrence of the species of Cladocera at the sampling sites. The first ordination axis, with its eigenvalue $\lambda = 0.292$, differentiated sampling sites and clearly identified the RXII core. The core was characterized by the highest species diversity of Cladocera compared to other studied cores.

The gradient represented by the second ordination axis, with its eigenvalue $\lambda = 0.195$, clearly grouped the cores in two groups: cores from ponds existing during mine exploitation, and ponds created after its closing. In the upper part of the diagram was placed RXX and RXXI cores (created after mining stopped). In this part of the diagram there are also other cores of the same group: RI, RII, RXVIII and RXIX. In reservoirs mentioned above, a small number of species and at low density was found (Figures 3 and 4).

The distribution of the species in the ordination space of the diagram shows that they occur in particular cores. *Daphnia cucullata*, *Alonella nana*, *Graptoleberis testudinaria* and *Alonella excisa* were located in the right part of the diagram and were associated with the RXII core (Figure 6).

**Table 3.** Pearson's correlation—significant relationships between heavy metals concentrations and density of Cladocera (A: UP—subsidence ponds formed after the closure of the mine; B: DOWN—subsidence ponds created during the peak of Zn and Pb ore exploitation).

| A Variables | Cu | Acroperus harpae | Alona sp. | Alona affinis | Alona quadrangularis | Chydorus sphaericus |
|---|---|---|---|---|---|---|
| Alona affinis | −0.53 p = 0.028 | | 0.94 p = 0.000 | | | |
| Alona guttata | −0.60 p = 0.01 | | 0.83 p = 0.000 | 0.84 p = 0.000 | | |
| Bosmina coregoni | | | | | 0.80 p = 0.000 | |
| Chydorus sphaericus | | 0.65 p = 0.004 | 0.62 p = 0.008 | 0.57 p = 0.018 | | |
| Daphnia sp. | −0.48 p = 0.049 | | | | | |
| Pleuroxus truncatus | | 0.65 p = 0.005 | | | | 0.82 p = 0.000 |

| B Variables | Zn | Pb | Cu | Alona sp. | Alona affinis | Alonella excisa | Alona quadrangularis | Alona rectangula | Alonella nana | Chydorus sphaericus | Eurycercus lamellatus | Pleuroxus truncatus |
|---|---|---|---|---|---|---|---|---|---|---|---|---|
| Alona sp. | −0.40 p = 0.042 | −0.41 p = 0.034 | −0.45 p = 0.021 | | | | | | | | | |
| Alona affinis | | | −0.48 p = 0.012 | 0.66 p = 0.00 | | | | | | | | |
| Alonella excisa | | | | 0.71 p = 0.00 | 0.93 p = 0.00 | | | | | | | |
| Alona guttata | | | | 0.64 p = 0.00 | | | | | | | | |
| Alona quadrangularis | | | −0.41 p = 0.037 | | | | | | | | | |
| Alona rectangula | | | | 0.73 p = 0.00 | 0.90 p = 0.00 | 0.97 p = 0.00 | | | | | | |
| Alonella nana | | | −0.40 p = 0.041 | | | | 0.99 p = 0.00 | | | | | |
| Daphnia cucullata | | | −0.41 p = 0.037 | | | | | | | 0.99 p = 0.00 | | |
| Eurycercus lamellatus | | | | | | | | | | 0.65 p = 0.000 | | |
| Graptoleberis testudinaria | | | −0.41 p = 0.037 | 0.69 p = 0.00 | 0.96 p = 0.00 | 0.98 p = 0.00 | | 0.95 p = 0.000 | | | | |
| Pleuroxus truncatus | | | | | | | | | | 0.44 p = 0.023 | 0.41 p = 0.039 | |
| Pleuroxus trigonellus | | | | | | | | | | 0.52 p = 0.006 | 0.43 p = 0.029 | 0.72 p = 0.000 |

Canonical correspondence analyses (CCA) showed the importance of heavy metal concentration in sediment cores on the distribution of Cladocera taxa. The results showed that the first axis explains 50.0%, and the second 21.9% of the variability of the occurrence of species. In the CCA ordination diagrams, the location of species in relation to axis I and II, as well as the intensity of changes of environmental variables are presented. The model of CCA is highly statistically significant (The Monte Carlo permutation test: First canonical axis F = 6.148, p = 0.002; all canonical axes F = 2.757, p = 0.002). The location of the species in the right part of the ordination diagram is clearly visible, which indicates their intolerance to higher concentrations of heavy metals (Cu and Cd) (Figure 7).

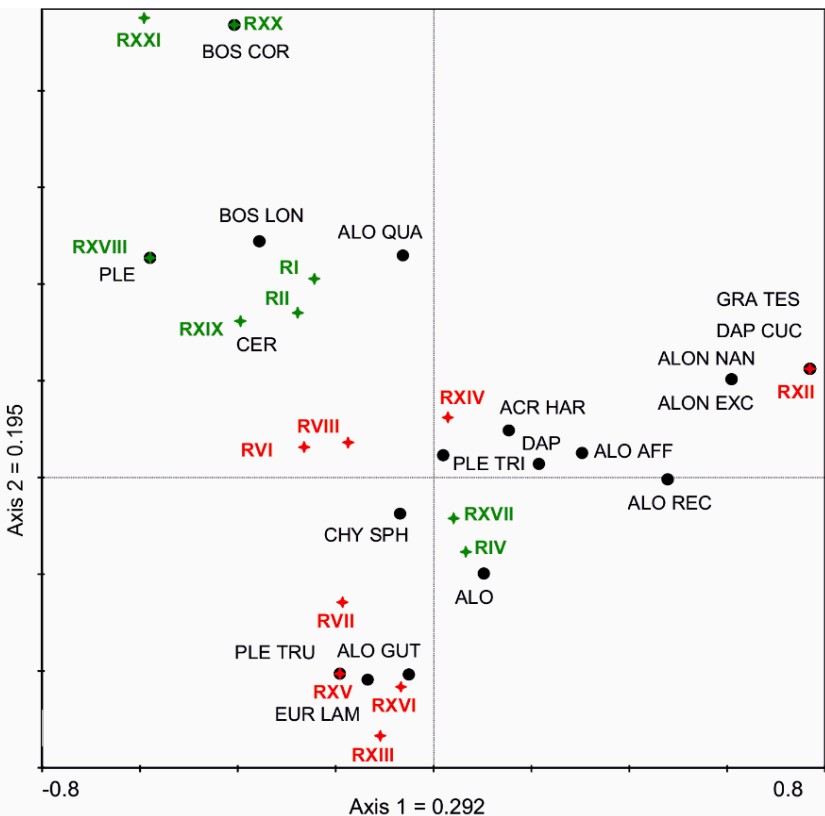

**Figure 6.** Correspondence analysis (CA) diagram for Cladocera fauna in sediment cores of subsidence ponds (ACR HAR: *Acroperus harpae*; ALO: *Alona* sp.; ALO AFF: *A. affinis*; ALO GUT: *A. guttata*; ALO QUA: *A. quadrangularis*; ALO REC: *A. rectangula*; ALON EXC: *Alonella excisa*; ALON NAN: *A. nana*; BOS COR: *Bosmina coregoni*; BOS LON: *B. longirostris*; CER: *Ceriodaphnia* sp.; CHY SPH: *Chydorus sphaericus*; DAP: *Daphnia* sp.; DAP CUC: *D. cucullata*; EUR LAM: *Eurycercus lamellatus*; GRA TES: *Graptoleberis testudinaria*; PLE: *Pleuroxus* sp.; PLE TRU: *P. truncatus*; PLE TRI: *P. trigonellus*). UP: Sediment cores are marked in green color; DOWN: Sediment cores are marked in a red color.

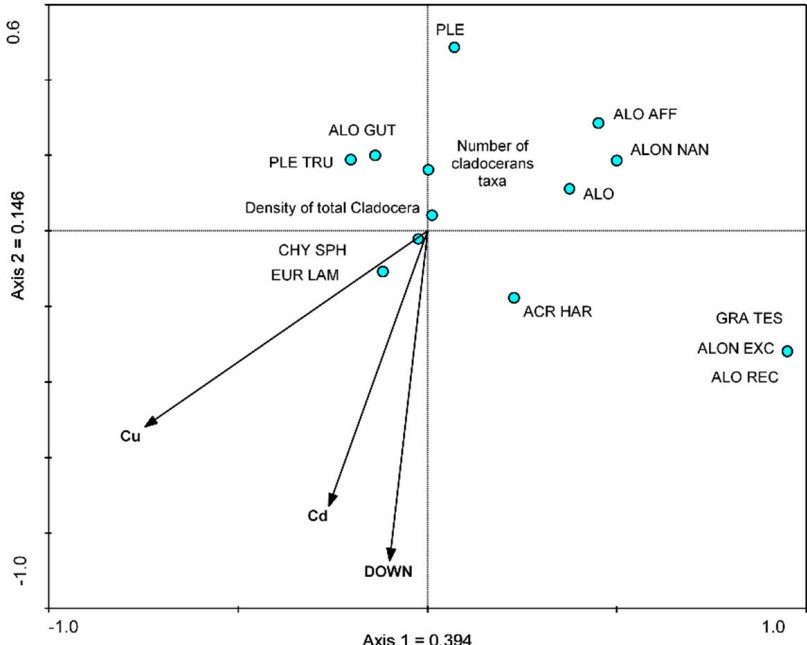

**Figure 7.** Canonical correspondence analysis (CCA) diagram for selected Cladocera species. The biplot illustrates the relationship between the heavy metal concentration in the sediment cores and Cladocera fauna. Taxa name code see Figure 6.

## 4. Discussion

The obtained results indicated strong contamination by Cd, Pb and Zn, and lower by Cu of sediment cores of UP and DOWN subsidence ponds of the Chechło River if compared to local geochemical background [8] and sediments of unpolluted areas [14]. The Cd, Pb and Zn concentrations were from tens to hundreds of times higher compared to those in the reference lake sediments (Figure 2) [14]. Such high heavy metals concentrations in the Chechło River sediments are typical for water ecosystems exposed to discharge from the Zn and Pb mines [6,15]. Metals discharged with mine waters and suspended sediments to the Chechło River during mine operation affected the DOWN subsidence ponds, what was reflected in maximal/very high Cd, Pb and Zn concentrations in the investigated sediment cores. The mine closure resulted in the drop of decrease in the river water contamination and lower metal concentrations [8] reflected in the large difference between contamination of the UP and DOWN subsidence ponds. Despite the progressive decrease of the most recent sediment contamination with Cd, Pb and Zn, their concentrations in all cores still exceed toxic levels for many aquatic organisms [16]. Moreover, Cu concentrations exceeded the threshold concentration (TEC, 86 µg g$^{-1}$;16) for 80% of the samples of the UP and DOWN subsidence ponds sediments. The sediment-associated heavy metals might be released from the sediment to the overlying water under changing physical-chemical conditions particularly as a result of sediment resuspension or oxidation of surface sediment strata during floods after heavy rainfalls or during the melting period [17,18].

Antropogenic pollution, such as heavy metal contamination, could be a detrimental and intense stressor of water ecosystems. The Cladocera are an important component of the zooplankton community in water bodies with both pelagic and littoral habitats. The Cladocera analysis raises a discussion about the influence of heavy metal pollution as a human activity. Cladocera assemblages from the subsidence ponds show significant differences in species composition, their densities and especially domination. In all subsidence ponds dominated *Ch. sphaericus* which is tolerant to water pollution and seems to be a 'specialist' in tolerance to a wide range of abiotic conditions [19]. However, some authors wrote that the long-term exposure of this species to Cu results in a reduction in the rate of population growth [7,20], but also Burton et al. [20] observed, that another species *Alona quadrangularis*

is restricted to sites of low Cu contamination. Our results confirmed that *Ch. sphaericus* tolerated high concentration of Cu, but *A. quadrangularis* is not tolerant of this metal (Table 3; Figure 7). In both types of subsidence ponds (UP and DOWN) high concentration of Cu was not tolerated by species of *Alona*, *Alonella*, *Daphnia* and *Graptoleberis*. Sarma et al. [21] showed no single species of studied Cladocera (e.g., *Alona rectangula*, *Daphnia laevis*, *D. pulex*, *D. similis*, *Ceriodaphnia dubia)* was consistently sensitive for stress from heavy metal. The sensitivity of the Cladocera to the concentration of heavy metals in the environment is also dependent on other abiotic factors. Additionally, it is possible that the temporary presence of *B. longirostris*, dominant presence of *Ch. sphaericus* and littoral cladocerans, e.g., *Alona affinis*, *A. rectangula*, which prefer more fertile water, reflect a response of this group to changes in the local habitat (macrophyte abundance with more food for the cladocerans) [22]. In the investigated subsidence ponds, we also observed preferences of cladoceran assemblages to stagnant water overgrown by macrophytes.

Cluster analysis divided our sampling cores into two groups—one with sediment less polluted by the heavy metals that accumulated after mine cessation, and a second group of highly polluted sediments by heavy metal that existed during mine operation (Figure 2; Figure 5). The impact of metals on the Cladocera assemblage is reflected in the Shannon index (Table 2), where we observe a greater number of cores with the value of the index above 1 (in reservoir formed after the mine was closed), which may mean recovery an increase in biological diversity of organisms and improvement of the quality of the environment.

The total abundance of Cladocera remains decreased dramatically in the postindustrial sediments (UP subsidence pond) compared with the industrial sediments (DOWN subsidence ponds). The subfossil records for all Cladocera taxa investigated very clearly revealed that the onset of industrial activities dramatically altered the ecology of small subsidence ponds which emerged in the river valley. Based on existing knowledge of aquatic ecosystems and their responses to metal pollution, we attempt to interpret the observed industrial to postindustrial changes or trends in biota in the studied ponds. Despite more than 10 years since of heavy metal from Pb-Zn ore mining cessation, we observed little evidence of recovery in Cladocera assemblages, which we observed in the value of Shannon diversity index, as well as the occurrence of *Daphnia* remains in RXII, the more polluted subsidence pond existing during mine working. Doig et al. [4] observed similar results in Ross Lake sediment impacted by mining, metallurgical and municipal activities where Cladocera remains have low abundance in postindustrial times and shows minimal signs of recovering.

## 5. Conclusions

Sediment cores of ponds on the floodplain of River Chechło allowed reconstruction of human impact. Cores of UP and DOWN ponds show various contamination by Pb, Zn, Cd and Cu, the reaction of Cladocera assemblages by changing plankton species composition and their density observed in sediments.

The analysis of Cladocera remains allowed to reconstruct pre-mining condition in the studied subsidence ponds and also showed the environmental conditions of the studied ponds when the Zn-Pb main was operating and after when was closed. The occurrence of different ecological groups of cladocerans (diversity in taxa and density) in studied subsidence ponds may suggest differences in ponds types in past and present, and showed important changes in water quality during mine operation and after it was closed. In the sediment core layers with lower metal concentrations we observed a little recovery of Cladocera assemblages. In all subsidence ponds *Ch. sphaericus* was the dominant species, and most abundant in studied sediments. The dominance of less sensitive species confirmed communities adapted to chronic high metal contamination.

The sediment quality assessment will help to understand persistent adverse effects of sediment contamination on aquatic communities and to better recognize factors decisive on mechanisms of the succession of aquatic organisms in contaminated river systems.

**Author Contributions:** A.P. and D.C. were responsible for the research design. A.P., A.Z.W., E.S.-G. and D.C. analyzed the data, prepared drafted the text and figures. A.C. performed statistical analyses. All authors participated in discussions and editing.

**Funding:** This research was funded by National Science Center grant n. 2014/15/B/ST10/03862 and under Institute of Nature Conversation, Polish Academy of Sciences subvention.

**Acknowledgments:** We would like to thank Hanna Kuciel for their help with drawing works. We extend our thanks to Henri Dumont for the assistance with our text and in language editing. The researches are a part of a grant titled: Reconstruction and prognosis of response of a river system affected by Pb-Zn ore extraction to mining cessation.

**Conflicts of Interest:** The authors declare no conflict of interest.

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
