# Peer review of "Response of Cladocera Fauna to Heavy Metal Pollution, Based on Sediments from Subsidence Ponds Downstream of a Mine Discharge (S. Poland)"

_water, doi:10.3390/w11040810_

Round 1
Reviewer 1 Report
Reviewer comments to the manuscript
The reviewed manuscript fills into the main trend of worldwide paleolimnological researches, it has also a new character, because of the testing Cladocera species environment relations to environmental variables (Pb, Zn, Cd, and Cu) and are giving a data about nowadays and past condition.
The article touches on the problem of pollution of the environment with heavy metals. Indicates their impact on the development of freshwater zooplankton. On the basis of Cladocera remains, included in the sediments, indicated the mine's impact during its operation and after its closing.
The manuscript, I read several times, contains many original and interesting data. The subject of the study is important. The research encloses interesting results of significant relationships between Cladocera species and heavy metals.
Authors were using some numerical methods, indicated significant correlations among the species composition, and the human impact (during mine activity and after closing).
The aim of the study is clearly stated, discussion and conclusions are short but clearly specified
I am recommending the paper, however some changing is required, before the manuscript can be published.
Comments:
Terminology:
1. The authors used different spellings for the same name. This should be standardized - in all pages in the text. Cladocera – capital letter, cladoceran – small letter
(For example on pages: Line: 19, 31, 37, 56, 62, 121, 182, etc. also in the fig. 6.
2. A shortcut to determine – individuals per 1 cm3 – should be ind. /1 cm3 (on many pages).
3. Line 137: not Ca
4. Line 157: which means – "with other reservoirs"
5. Line 164 - vegetation on Fig. 1.?? there are not in the legend
6. Line 174 - six not seven taxa?
7. Line 177 - “only dominated littoral taxa”? - In all profiles dominated littoral species. Not only in RXXI – were dominated Bosminidae. Probably authors have in mind the highest number of littoral taxa.
Chapters:
Abstract
Line 27 – 3 taxa - Alona sp. Is not taxa, you can write other taxa of Alona?
Materials and Methods
The description must be precise and complete. The authors are referring to the work of Ciszewski (2019), in which there are some more accurate data, but the reader who reads this (reviewed) article should have enough information to understand the methodological work. Lack of description of lithology and exact methods of sediment sampling, causes misunderstanding of methods used. The description of the chemical method should include the cited literature by which method was taken. Lack of description about the time when the mine was in operation and when it was closed results in a reservation as to the reliability of the results obtained. A reviewer familiar with the problem is convinced of the correct results, but such a belief must also have the reader of this article.
Tables:
Table 1. Caption of the table is not precisely – it gives info. About lowers and higher concentration (ranged).
Signature under the table incorrect = Reference lake sediment [14]
Table 2.
Caption in the table - better: Range (min.-max.) of total Cladocera individuals per 1 cm3
(Not clear – depends from the layer?)
Table 4.
The organization of the table not clear, Alona sp. (other Alona?), the names of species – consequently in two row, the table should be more compact, etc.
Figures:
Fig.1.
Complete the description under Fig.1. – RI – RXXII – cores (profiles?), how the valley range has been marked, what about macrophytes?
Fig. 2 and 3.
Should be rewritten.:
- Are not clear. No time scale, huge difference of scale for individuals -leads to halos,
- was there not third layer in RII?
Alona sp. ?? Daphnia sp., ??
Cladocera nd -?, (This is probable mean a lack of density of Cladocera)? – better – no Cladocera, but if it is the way, it does not match - for example in RXII, RVII
Fig. 4 too big,
Fig. 5. The caption for the figure must be described more, included some info from the text - what it shows. Axis 1 and axis 2 must be marked on the figure. Which means: "in our water bodies" – it must be clear that the data are from sediments.
Fig. 6. The caption for the figure must be described more, included some info from the text - what it shows. Axis 1 and axis 2 must be marked on the figure. The biplot illustrating the relationship between ………
Code for the names of the Cladocera cancel – (write) species name code see: fig. 5.
In the fig.: Cladocera - Capital letter, down – should be DOWN, ALO (others)?
References
Some letters mistake, points, etc. – check.
Author Response
Reviewer 1
Answer:
Authors would like to thanks for Reviewer for the essential comments which improve manuscript.
“Comments:
Terminology:
1. The authors used different spellings for the same name. This should be standardized - in all pages in the text. Cladocera – capital letter, cladoceran – small letter
(For example on pages: Line: 19, 31, 37, 56, 62, 121, 182, etc. also in the fig. 6.”
Answer: We improved all suggestions mistakes in whole text.
“2. A shortcut to determine – individuals per 1 cm3 – should be ind. /1 cm3 (on many pages).”
Answer: A shourtcut is improved in whole text. Individuals per 1 cm3 we can improved as ind. /1 cm3 , but we used as ind. cm -3 as usual, it is used in hydrobiological studies.
“3. Line 137: not Ca
4. Line 157: which means – "with other reservoirs"
5. Line 164 - vegetation on Fig. 1.?? there are not in the legend
6. Line 174 - six not seven taxa?
7. Line 177 - “only dominated littoral taxa”? - In all profiles dominated littoral species. Not only in RXXI – were dominated Bosminidae. Probably authors have in mind the highest number of littoral taxa.”
Answer: All points (3-7) are improved in the text.
“ Chapters:
Abstract
Line 27 – 3 taxa - Alona sp. Is not taxa, you can write other taxa of Alona?”
Answer: It is improved.
“Materials and Methods
The description must be precise and complete. The authors are referring to the work of Ciszewski (2019), in which there are some more accurate data, but the reader who reads this (reviewed) article should have enough information to understand the methodological work. Lack of description of lithology and exact methods of sediment sampling, causes misunderstanding of methods used. The description of the chemical method should include the cited literature by which method was taken. Lack of description about the time when the mine was in operation and when it was closed results in a reservation as to the reliability of the results obtained. A reviewer familiar with the problem is convinced of the correct results, but such a belief must also have the reader of this article.”
Answer: In material and methods we added some information about lithology and sediment sampling and also we cited a literature by which method was taken for chemical method.
Authors also added some information in introduction section about when the mine was in operation and when it was closed and added literature.
“Tables:
Table 1. Caption of the table is not precisely – it gives info. About lowers and higher concentration (ranged).
Signature under the table incorrect = Reference lake sediment [14]”
Answer: Table 1 was deleted and changed on Figure 2. The Rewiever 3 also suggested that changes and improvement connected with this table. Now the concentrations of heavy metals is presented in section of sediment cores.
“Table 2.
Caption in the table - better: Range (min.-max.) of total Cladocera individuals per 1 cm3
(Not clear – depends from the layer?)
Table 4.
The organization of the table not clear, Alona sp. (other Alona?), the names of species – consequently in two row, the table should be more compact, etc.”
Answer: Table 2 (present is Table 1) and 4 (present is Table 3) were improved.
“Figures:
Fig.1.
Complete the description under Fig.1. – RI – RXXII – cores (profiles?), how the valley range has been marked, what about macrophytes?”
Answer: Figure 1 was improved.
“Fig. 2 and 3.
Should be rewritten.:
- Are not clear. No time scale, huge difference of scale for individuals -leads to halos,
- was there not third layer in RII?
Alona sp. ?? Daphnia sp., ??
Cladocera nd -?, (This is probable mean a lack of density of Cladocera)? – better – no Cladocera, but if it is the way, it does not match - for example in RXII, RVII “
Answer: Figure 2 and 3 were rewritten. Present are Figure 3 and Figure 4, respectively. The signature “Cladocera nd “was deleted because it was mistake.
“Fig. 4 too big,
Fig. 5. The caption for the figure must be described more, included some info from the text - what it shows. Axis 1 and axis 2 must be marked on the figure. Which means: "in our water bodies" – it must be clear that the data are from sediments.
Fig. 6. The caption for the figure must be described more, included some info from the text - what it shows. Axis 1 and axis 2 must be marked on the figure. The biplot illustrating the relationship between ………
Code for the names of the Cladocera cancel – (write) species name code see: fig. 5.
In the fig.: Cladocera - Capital letter, down – should be DOWN, ALO (others)?”
Answer: All figures from 4 to 6 were improved. Present are Figures from 4 to 7.
“References
Some letters mistake, points, etc. – check.”
Answer: In reference chapter all mistakes were improved.

Reviewer 2 Report
In my opinion, the paper is very good and can be published as it is. The problem is important, research methods are adequate and conclusions are sound. Recovery of communities after polution is an important topic of modern research, and Cladocera are good model organisms to study historical changes.
Author Response
Reviewer 2
“In my opinion, the paper is very good and can be published as it is. The problem is important, research methods are adequate and conclusions are sound. Recovery of communities after polution is an important topic of modern research, and Cladocera are good model organisms to study historical changes. “
Answer:
Authors would like to thanks for very good review. According for suggestions of Reviewers 1 and 3 the manuscript was slightly improved

Reviewer 3 Report
Presented paper “Response of Subfossil Cladoceran remains to environmental change in subsidence ponds downstream of a heavy metal mine discharge in southern Poland.” is presenting the results of geochemical and Cladocera analyses from small polluted subsidence lake in the S Poland.
In my opinion the subject of paper is important for understanding reaction of zooplankton community to water heavy metal pollution. Authors are showing an influence of heavy metal pollution to the Cladocera communities during mining operation and after time when mine was close. An important is that paper is presenting respond of zooplankton to this factor and showing recuperation of Cladocera with the imporooving water environment after mine was closed. Reconstruction is based on lacustrine sediments. Up to now only few studies in the area were made. Presented paper is showing relationship between Cladocera community and heavy metal content in the lake waters, so this new data has important value. The results of this study can help us to better understand processes which are occurring in the polluted reservoirs and as well presented result will be useful for paleolake reconstruction of heavy metal content.
However, in spite of this that subject of the paper is important, presented manuscript in my opinion, needs a major revision. I strongly believe that paper after improving and correction is going to be an important paper for limnology and especial for paleolimnology (tracking heavy metals pollution in the lakes sediment). The biggest improvement is required in the materials and methods section
Below I point out suggestion to some part of article:
Materials and methods section requires supplements and explanations.
1) Please provide an reference of the time for subsidence point which were created after and before mining operation
2) On the figure one are more sites (20) then used in the studies (please remove those which were no used). Also will be easier for reader, in my option, when you will slightly rename sampling point e.g. RI for UPRI or RV for DOWNRV – this will make all article more easy to follow.
3) It is necessary to describe and present how sediment cores were divided in section and please provide the length (cm) of each studied core
4) Please provide how you take samples for Cladocera analysis – e.g. dry wet sediments, did you take 1cm3 of homogenized sediment from section or ….?
5) In my opinion Table 1 and Fig 2 and 3 have to be rewritten in the presented state are difficult to read and made confusion. My suggestion are following: Table one It can be replace by depth diagram of heavy metal content in each study site (R1-RXVI). Figure 2 and 3 – is necessary to present each results of subfossil Cladocera analysis – from text I know that in the lower section of sediment core there were not Cladocera – however in the fig 2 and 3 – I am not able to observe it. Maybe it is also good idea to present those results in depth diagram. Please explain also what’s mean on diagram Cladocera nd. And maybe word density will be better to replace by Number of Cladocera individuals in the 1cm of fresh/ dry sediment.
6) Table 4 – some graphical changes are required
In my opinion others part of manuscript, like second part of results (Fig 4, 5, 6) as well discussion and conclusion are well organize and presented. Those part are informative and are bringing new information about relationship between Cladocera communities and heavy metal pollution.
I think that presented manuscript is touching an important environmental observation and after improving first part of results will be one of the most significant paper in the field of reconstruction form lacustrine sediment of heavy metals influence on Cladocera communities.
Edyta Zawisza
Author Response
Reviewer 3
Answer:
Authors would like to thanks for Reviewer for the essential comments which improve manuscript.
“Below I point out suggestion to some part of article:
Materials and methods section requires supplements and explanations.
1) Please provide an reference of the time for subsidence point which were created after and before mining operation”
Answer: Authors added some information in introduction section about when the mine was in operation and when it was closed and added literature.
“2) On the figure one are more sites (20) then used in the studies (please remove those which were no used). Also will be easier for reader, in my option, when you will slightly rename sampling point e.g. RI for UPRI or RV for DOWNRV – this will make all article more easy to follow.”
Answer: The figure 1 was improved and mistake cores were removed. We tried rename sampling point for longer name as suggesting reviewer but the created names were so long and not readable in figures and tables. Authors thinks that presented name of sampling point divided on two groups UP and DOWN are much readable than reviewer suggested.
“3) It is necessary to describe and present how sediment cores were divided in section and please provide the length (cm) of each studied core”
Answer: In material and methods we added information about how were divided sediment cores and how were sampling. In figures it was showed the length of each studied cores .
“4) Please provide how you take samples for Cladocera analysis – e.g. dry wet sediments, did you take 25px3 of homogenized sediment from section or ….? “
Answer: In material and methods in Cladocera sediment preparation paragraph was added information how Cladocera samples were provided.
“5) In my opinion Table 1 and Fig 2 and 3 have to be rewritten in the presented state are difficult to read and made confusion. My suggestion are following: Table one It can be replace by depth diagram of heavy metal content in each study site (R1-RXVI). Figure 2 and 3 – is necessary to present each results of subfossil Cladocera analysis – from text I know that in the lower section of sediment core there were not Cladocera – however in the fig 2 and 3 – I am not able to observe it. Maybe it is also good idea to present those results in depth diagram. Please explain also what’s mean on diagram Cladocera nd. And maybe word density will be better to replace by Number of Cladocera individuals in the 25px of fresh/ dry sediment.”
Answer: According to Reviewer sugesttions we changed Table 1 for Figure 2 and Figure 2 and 3 rewritten (present Figure 3 and 4) to more readable figures and presented in depth diagram.
Cladocera nd it is our mistake and we delayed it from figures, it means Cladocera no determinate.
We live word density because it means the same number of individuals per unit of area or volume.
“6) Table 4 – some graphical changes are required”
Answer: The table 4 was improved. Present is Table 3.

Round 2
Reviewer 1 Report
Reviewer comments to the rewritten manuscript
The manuscript, was rewritten. The authors have taken my remarks into account. However, the manuscript still need to make a minor correction.
The tittle of the manuscript is not precise (subfossil = remains, response of remains?)
Comments:
Terminology and small errors:
1. sediment stratigraphy - should be better (more precisely) sediment lithology (p.90,96 and other)
2. A shortcut to determine – individuals per 1 cm3– should be ind. /1 cm3 (on many pages, also in table and figures). In the text is: cm -3: it cannot be minus, regarding to the description of the method
3. Line 111: “and the percentages were calculated from the total number individuals” leave this sentence
4. Line 173: Table 1. Number of taxa and range (min.-max.) of total Cladocera specimens in cores of subsidence ponds.
5. Line 177: Total Cladocera individuals in …. (remains not individuals)
6. Line 193, 203 Fig. 3 and 4 - description should be more precise
7. Line 246 – cyt. Fig. 3, 4 (not 5)
8. Line 278 - cyt Fig. 6 (the code is on fig. 6)
Fig.4 – is to small - maybe it can be presented in a horizontal position?
Author Response
Reviewer 1
Answer:
Authors would like to thanks for Reviewer for the essential comments which improve manuscript.
“ The tittle of the manuscript is not precise (subfossil = remains, response of remains?)”
Answer: We improved title according with suggestion of Rewiever 3, now is: Response of Cladocera fauna to heavy metal pollution, based on sediments from subsidence ponds downstream of a mine discharge (S Poland).
“Comments:
Terminology and small errors:
1. sediment stratigraphy - should be better (more precisely) sediment lithology (p.90,96 and other)
2. A shortcut to determine – individuals per 1 cm3– should be ind. /1 cm3 (on many pages, also in table and figures). In the text is: cm -3: it cannot be minus, regarding to the description of the method
3. Line 111: “and the percentages were calculated from the total number individuals” leave this sentence
4. Line 173: Table 1. Number of taxa and range (min.-max.) of total Cladocera specimens in cores of subsidence ponds.
5. Line 177: Total Cladocera individuals in …. (remains not individuals)
6. Line 193, 203 Fig. 3 and 4 - description should be more precise
7. Line 246 – cyt. Fig. 3, 4 (not 5)
8. Line 278 - cyt Fig. 6 (the code is on fig. 6)
Fig.4 – is to small - maybe it can be presented in a horizontal position? “
Answer: All points (1-8) are improved in the text, figures and table 1.

Reviewer 3 Report
Presented paper “Response of Subfossil Cladoceran remains to environmental change in subsidence ponds downstream of a heavy metal mine discharge in southern Poland.” is presenting the results of geochemical and Cladocera analyses from small polluted subsidence lake in the S Poland.
I received presented paper second time, after authors revision. In my opinion the subject of paper is important for understanding reaction of zooplankton community to water heavy metal pollution. Authors improvement of paper is in most cases fulfilled my expectations. However, below I am pointing out some small required changes.
I strongly believe that paper after improving is going to be an important paper for limnology and especial for paleolimnology (tracking heavy metals pollution in the lakes sediment).
Suggestion:
Title is unclear. I suggest following (or something similar): Response of Cladocera fauna to heavy metal pollution, based on sediments from subsidence ponds downstream of a mine discharge (S Poland).
In all text authors are using indNaN-3 – I think that authors are thinking individuals per 1 cm3
-line 90 and 96, replace word stratigraphy by lithology
-Line 112 – if really percentage was calculated ? – in the text authors are using density – its mean individuals per 1 cm3 please clarify this part of the text
-Line 177 ……. Total density of Cladocera remains ??? I think authors are thinking here individuals
Table 1
name of the table is not suitable with table content (in the table description is total density in cores but range is in 25px3 (not in cores)) I propose to change both table description and name of the column. I propose following name of column: Range (min.-max) of Cladocera individuals in 1 cm3of sediment. Also table 1 is in contrary in some places with figure 3. e.g. in Table 1 is written that in point RIV range is 4-17 but in Figure 3 in the bottom layer is more individuals than 17 (Alona sp. 7, Alona affinis 5, Alona guttata -2, Alona quadrangularis 1, Chydorus sphaericus – 7 ). Please check carefully both table 1 and figures 3 and 4
I am also suggesting to change title of Figure 3 and 4 for following: Diagram of the absolute number of Cladocera individuals in 1 cm3 of sediment from
Fig 4 – is to small – I propose to make it horizontal
- Line 246 – Instead of Fig 5 should be 3 and 4
- Line 278 Instead of Fig 5 should 6
Edyta Zawisza
Author Response
Reviewer 3
Answer:
Authors would like to thanks for Reviewer for the essential comments which improve manuscript.
“ Comments and Suggestions for Authors
Title is unclear. I suggest following (or something similar): Response of Cladocera fauna to heavy metal pollution, based on sediments from subsidence ponds downstream of a mine discharge (S Poland).”
Answer: We improved title according with Rewiever suggestion.
“In all text authors are using indNaN-3 – I think that authors are thinking individuals per 1 cm3
-line 90 and 96, replace word stratigraphy by lithology
-Line 112 – if really percentage was calculated ? – in the text authors are using density – its mean individuals per 1 cm3 please clarify this part of the text
-Line 177 ……. Total density of Cladocera remains ??? I think authors are thinking here individuals
Table 1
name of the table is not suitable with table content (in the table description is total density in cores but range is in 25px3 (not in cores)) I propose to change both table description and name of the column. I propose following name of column: Range (min.-max) of Cladocera individuals in 1 cm3of sediment. Also table 1 is in contrary in some places with figure 3. e.g. in Table 1 is written that in point RIV range is 4-17 but in Figure 3 in the bottom layer is more individuals than 17 (Alona sp. 7, Alona affinis 5, Alona guttata -2, Alona quadrangularis 1, Chydorus sphaericus – 7 ). Please check carefully both table 1 and figures 3 and 4
I am also suggesting to change title of Figure 3 and 4 for following: Diagram of the absolute number of Cladocera individuals in 1 cm3 of sediment from
Fig 4 – is to small – I propose to make it horizontal
- Line 246 – Instead of Fig 5 should be 3 and 4
- Line 278 Instead of Fig 5 should 6 “
Answer: All comments and suggestions are improved in the text. We are cordially grateful for deeply analyzes. Now manuscript looks like much more better.
